# Many lncRNAs, 5'UTRs, and pseudogenes are translated and some are likely to express functional proteins

Zhe Ji[1,2†], Ruisheng Song[1†], Aviv Regev[2,3], Kevin Struhl[1]*

[1]Department of Biological Chemistry and Molecular Pharmacology, Harvard Medical School, Boston, United States; [2]Broad Institute of MIT and Harvard, Cambridge, United States; [3]Department of Biology, Howard Hughes Medical Institute, Massachusetts Institute of Technology, Cambridge, United States

**Abstract** Using a new bioinformatic method to analyze ribosome profiling data, we show that 40% of lncRNAs and pseudogene RNAs expressed in human cells are translated. In addition, ~35% of mRNA coding genes are translated upstream of the primary protein-coding region (uORFs) and 4% are translated downstream (dORFs). Translated lncRNAs preferentially localize in the cytoplasm, whereas untranslated lncRNAs preferentially localize in the nucleus. The translation efficiency of cytoplasmic lncRNAs is nearly comparable to that of mRNAs, suggesting that cytoplasmic lncRNAs are engaged by the ribosome and translated. While most peptides generated from lncRNAs may be highly unstable byproducts without function, ~9% of the peptides are conserved in ORFs in mouse transcripts, as are 74% of pseudogene peptides, 24% of uORF peptides and 32% of dORF peptides. Analyses of synonymous and nonsynonymous substitution rates of these conserved peptides show that some are under stabilizing selection, suggesting potential functional importance.

*For correspondence: kevin@ hms.harvard.edu

†These authors contributed equally to this work

## Introduction

In the central dogma, mRNAs are translated into proteins that carry out biological functions. On a genomic scale, translated regions are identified as open reading frames (ORFs) that are longer (typically >100 amino acids) than expected by chance, given sequence composition. In addition to mRNAs, mammalian cells contain other RNA transcripts generated by RNA polymerase II that are polyadenylated, spliced, and capped, but may not code for protein. One category consists of thousands of long RNAs that lack long open reading frames and have been considered to be non-coding (*Guttman et al., 2009*; *2010*; *Trapnell et al., 2010*; *Cabili et al., 2011*). A few lncRNAs play key regulatory roles in various biological processes via functional RNA domains that regulate chromatin modifications, DNA transcription, mRNA stability, and translation (*Rinn and Chang, 2012*; *Batista and Chang, 2013*; *Ulitsky and Bartel, 2013*). However, the biological functions of most lncRNAs remain unknown.

The human genome also encodes thousands of pseudogenes, which are homologous to protein-coding genes but have lost their coding ability and/or are not expressed (*Vanin, 1985*). Pseudogenes can function as competing endogenous RNAs (ceRNAs) regulating other RNA transcripts by competing for microRNAs (*Salmena et al., 2011*). Some pseudogenes are differentially expressed in human cancers (*Kalyana-Sundaram et al., 2012*; *Han et al., 2014*), but it is unknown if the RNAs expressed from pseudogenes are translated or have biological functions.

By definition, noncoding RNAs should not be translated into protein, but this can be difficult to ascertain using informatics alone because they contain short open reading frames that could be

**eLife digest** Our genes encode the instructions needed to make proteins. When a gene is switched on, it's DNA is used as a template to make molecules of messenger ribonucleic acid (RNA). These RNAs are then "translated" into proteins by large cell machines called ribosomes. Within the messenger RNA, a long region called an "open reading frame" is the section that encodes the protein.

The human genome also contains a vast amount of DNA that is not part of any gene. Cells can produce molecules of RNA from this DNA (so-called "non-coding RNAs"), but these RNAs are not thought to code for proteins because they lack long open reading frames. Non-coding RNAs can also be made from sections of DNA called "pseudogenes", which have lost their ability to code for proteins over the course of evolution. Furthermore, messenger RNAs also contain short open reading frames in the "untranslated" regions that flank the protein-coding region.

The extent to which cells translate non-coding RNAs to produce small proteins (or peptides) is not known. "Ribosome profiling" is a powerful method to determine which RNAs are translated, but it is not always possible to distinguish between the RNAs that are genuinely translated and those that just happen to be bound to ribosomes. Ji et al. overcome these limitations by developing a new computational method to analyse data from ribosome profiling.

The experiments show that thousands of non-coding RNAs in the human genome are, in fact, translated. This is many more than anticipated and represents approximately 40% of the lncRNAs and pseudogene RNAs, and 35% of untranslated regions in messenger RNAs. Ji et al. also found that a small group of all the lncRNA peptides in the human genome appear to have changed little over the course of evolution, which strongly suggests that they have specific roles in cells. The next challenge is to find out what roles the peptides encoded by these lncRNAs play in cells.

potentially translated. Even if a peptide is expressed from a putative non-coding RNA, it is difficult to determine whether the peptide has a biological function or is a mere by-product of an RNA that performs the biological function. However, there are a few examples of lncRNAs that are in fact translated into short peptides with biological roles (*Galindo et al., 2007*; *Kondo et al., 2010*; *Magny et al., 2013*; *Pauli et al., 2014*).

In addition, a number of mammalian mRNAs contain so-called 5' untranslated regions (5'UTRs) with one or more ORFs upstream of their canonical protein-coding regions (uORFs). Due to the scanning mechanism for translational initiation in which ribosomes scan in a 5' to 3' direction from the mRNA cap to find an initiation codon (*Sonenberg and Hinnebusch, 2009*), uORFs have the potential to regulate translation of the primary protein-coding ORF (*Calvo et al., 2009*; *Barbosa et al., 2013*). For example, translation of the uORFs in the yeast *GCN4* gene strongly inhibits translation of Gcn4 under normal conditions (*Hinnebusch, 2005*). However, during amino acid starvation, ribosomes reinitiate translation at the canonical AUG codon, thereby permitting increased synthesis of Gcn4 (*Hinnebusch, 2005*). In human cells, bioinformatic analyses and limited functional testing indicate that uORFs can inhibit protein production, but genome-wide functional analysis has yet to be performed (*Calvo et al., 2009*; *Barbosa et al., 2013*).

Ribosome profiling, the sequencing of ribosome-associated RNAs, represents a powerful assay for assessing translation in vivo in an unbiased manner on a genome-wide scale (*Ingolia et al., 2009*; *Ingolia, 2014*). In particular, ribosome profiling in mammalian cells reveals many reads derived from lncRNAs and 5' UTRs, and lncRNAs and 5'UTRs can be co-purified with 80S ribosome, indicating that these transcripts are translated (*Ingolia et al., 2011*; *2014*). However, unlike canonical protein coding-genes translated from mRNAs, many lncRNAs do not have a predominant ORF based on the ribosome release or disengagement scores (*Chew et al., 2013*; *Guttman et al., 2013*). However, due to a variety of limitations, previous analyses typically did not explicitly identify in-frame translated ORFs, and they identified only several hundred translated regions that do not correspond to canonical protein-coding regions. Importantly, ribosome profiling reads do not necessarily represent its active translation, due to potential artifacts from non-ribosomal entities and scanning ribosomes (*Guttman et al., 2013*; *Ingolia et al., 2014*).

Systematic examination of translation requires a computational method to identify *bona fide* translated ORFs in an unbiased fashion. Here we develop a method, RibORF, to analyze ribosomal profiling data and identify translated ORFs that combines alignment of ribosomal A-sites, 3-nt periodicity, and uniformity across codons. RibORF can effectively distinguish in-frame ORFs from overlapping off-frame ORFs, and it can distinguish reads arising from RNAs that are not associated with ribosomes. Using RibORF, we identify thousands of translated ORFs in lncRNAs, pseudogenes, and mRNA regions upstream (5'UTRs) and downstream (3'UTRs) of protein-coding sequences. Our results suggest that cytoplasmic noncoding RNAs are translated, and that some of these translated products are likely to be biologically meaningful based on their evolutionary conservation.

## Results

### Ribosome profiling experiment reveals in vivo translation in single nucleotide resolution

We performed ribosome profiling (*Figure 1A*) in two isogenic human cancer cell models: a Src-inducible mammary epithelial model and a Ras-dependent fibroblast model (*Hirsch et al., 2010*). Cells were treated either with cycloheximide, which inhibits translational elongation of ribosomes throughout the mRNA coding region, or harringtonine, which traps the ribosome at the site of translational initiation. After removing reads aligned to rRNAs and multiple genomic locations, we generated 44.0 and 21.2 million unique mappable reads upon cycloheximide treatment for breast epithelial and fibroblast cell transformation models, respectively. For harringtonine treatment, we obtained 5.9 and 9.0 million unique mappable reads for breast epithelial and fibroblast cells, respectively.

The length of ribosome-protected fragments (RPFs) ranges primarily between 24–31 nts (*Figure 1—figure supplement 1A*). Notably, RPFs with different length have variable distances between the 5' end and the ribosome A-site, as defined by canonical ORFs in protein-coding genes (*Figure 1—figure supplement 1B*). We used these offset distances in known protein-coding genes to account for the read length distribution and thereby align RPFs to specific A-site nucleotides throughout the entire dataset. Most expressed protein-coding ORFs show a clear 3-nt periodicity corresponding to *codon* triplets (*Figure 1B,C*). The 1st nucleotides of codons in an ORF contain about 65% of reads, while the 2nd and 3rd have 24% and 11%, respectively (*Figure 1B,C*). In addition, reads in protein-coding genes are uniformly distributed across codons in an ORF (*Figure 1D*). 73% of ribosome profiling reads map to canonical ORFs of mRNAs. 2% and 4% map to 5'UTRs and 3'UTRs of mRNAs, respectively, and 9% map to lncRNAs and pseudogenes, suggesting pervasive non-canonical translation (*Figure 1E*).

### Removing sequence reads that are not derived from translated RNA

Consistent with previous reports (*Ingolia et al., 2011*; *Guttman et al., 2013*), some ribosome profiling reads map to short noncoding RNAs, including small nucleolar RNA (snoRNA). As snoRNAs are located in nucleus, they should not be accessible to translation machinery located in cytoplasm. Indeed, the sequence reads in the snoRNAs map to a very narrow region and are comparable in the cycloheximide- and harringtonine-treated samples (*Figure 1F*), indicating they do not represent translated regions of these RNAs. To exclude reads that do not represent active translation, we developed a Percentage of Maximum Entropy (PME) approach to measure the uniformity of read distribution across codons in a candidate ORF (See Experimental Procedures). A PME value of 1 represents uniform read distribution, indicative of real translation, while smaller values indicate skewed distribution with a minimum value of 0 indicating reads at a single location, expected for reads not derived from translated RNA. As expected, candidate ORFs from short noncoding RNAs show drastically lower PME values, as compared to canonical protein coding ORFs (*Figure 1G*). Low PME values indicate RNAs that are not translated, but rather are protected in non-ribosomal protein complexes (*Ji et al., 2015*).

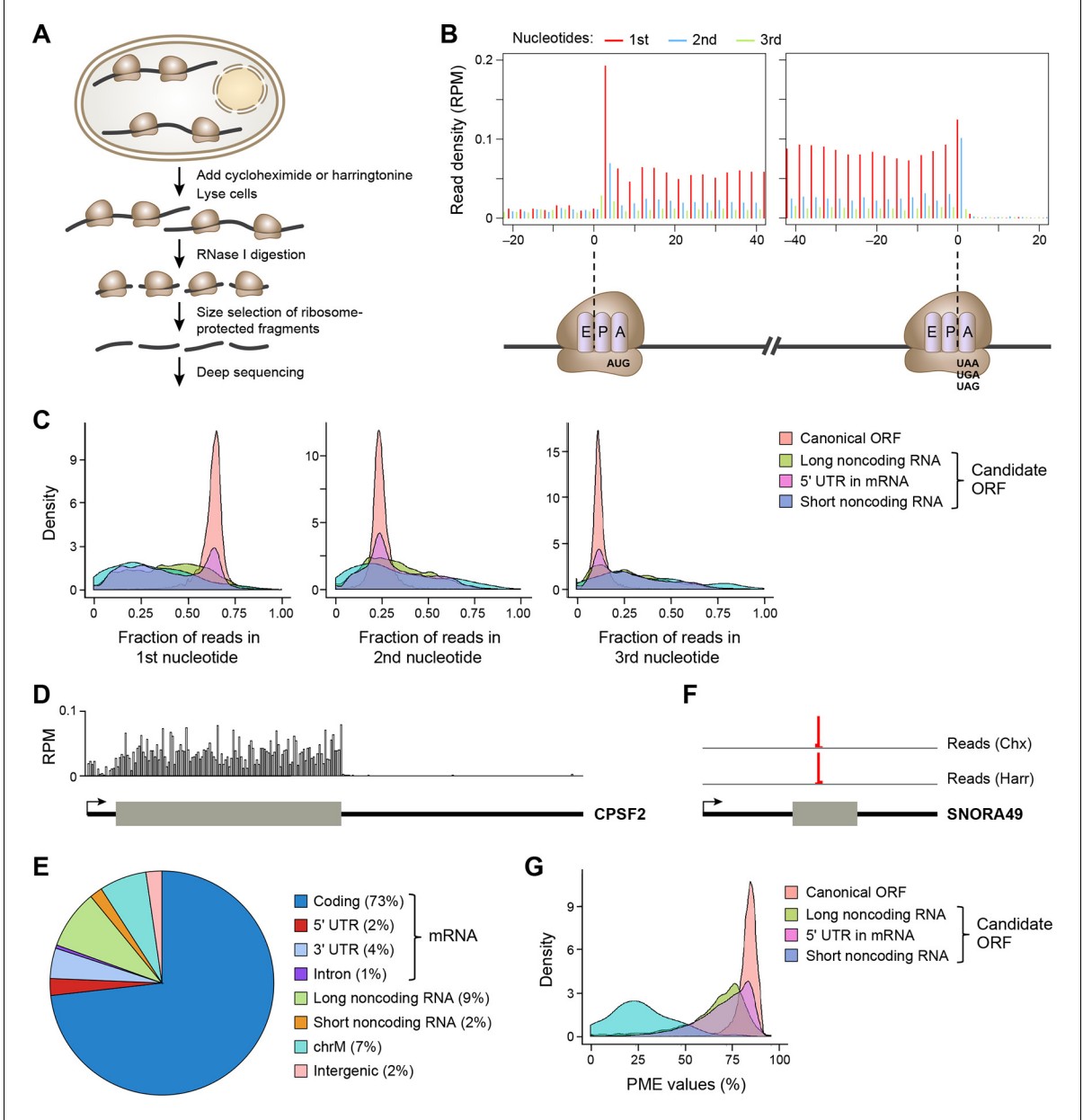

**Figure 1.** Ribosome profiling reveals in vivo translation with single nucleotide resolution. (**A**) Ribosome profiling experiment. (**B**) Read distribution (reads/million mappable reads; RPM) around start and stop codons of canonical protein coding genes. (**C**) Fractions of reads in 1st, 2nd and 3rd nucleotides of codons in the indicated types of ORFs. (**D**) Read distribution in the protein-coding gene CPSF2. The RPM value was calculated for every 20-nt region along the transcript. (**E**) Distribution of reads across human genome. (**F**) Read distribution of the snoRNA gene SNORA49 in cells treated with cycloheximide (Chx) or harringtonine (Harr). (**G**) Distribution of PME values in the indicated types of ORFs.

The following figure supplement is available for figure 1:

**Figure supplement 1.** Ribosome profiling data.

## RibORF identifies a large number of translated ORFs in lncRNAs, pseudogenes, and UTRs of mRNAs

Based on the 3-nt periodicity (*Figure 1C*) and uniformity of read distribution across codons (*Figure 1G*) of translated regions, we developed a Support Vector Machine classifier, RibORF, to identify translated ORFs from ribosome profiling data. The model was trained by using canonical

protein-coding ORFs as positive examples and off-frame ORFs from protein-coding regions and candidate ORFs from short noncoding RNAs as negative examples. The classifier using both features performed almost perfectly to separate positive and negative examples in a testing set (Area Under the ROC Curve [AUC] = 0.996), with 3-nt periodicity making a greater contribution (*Figure 2A*). The algorithm performed well for genes expressed at various levels, with AUC values greater than 0.993 for ORFs with RPKM > 1 (*Figure 2—figure supplement 1A*). In addition, the predicted translation

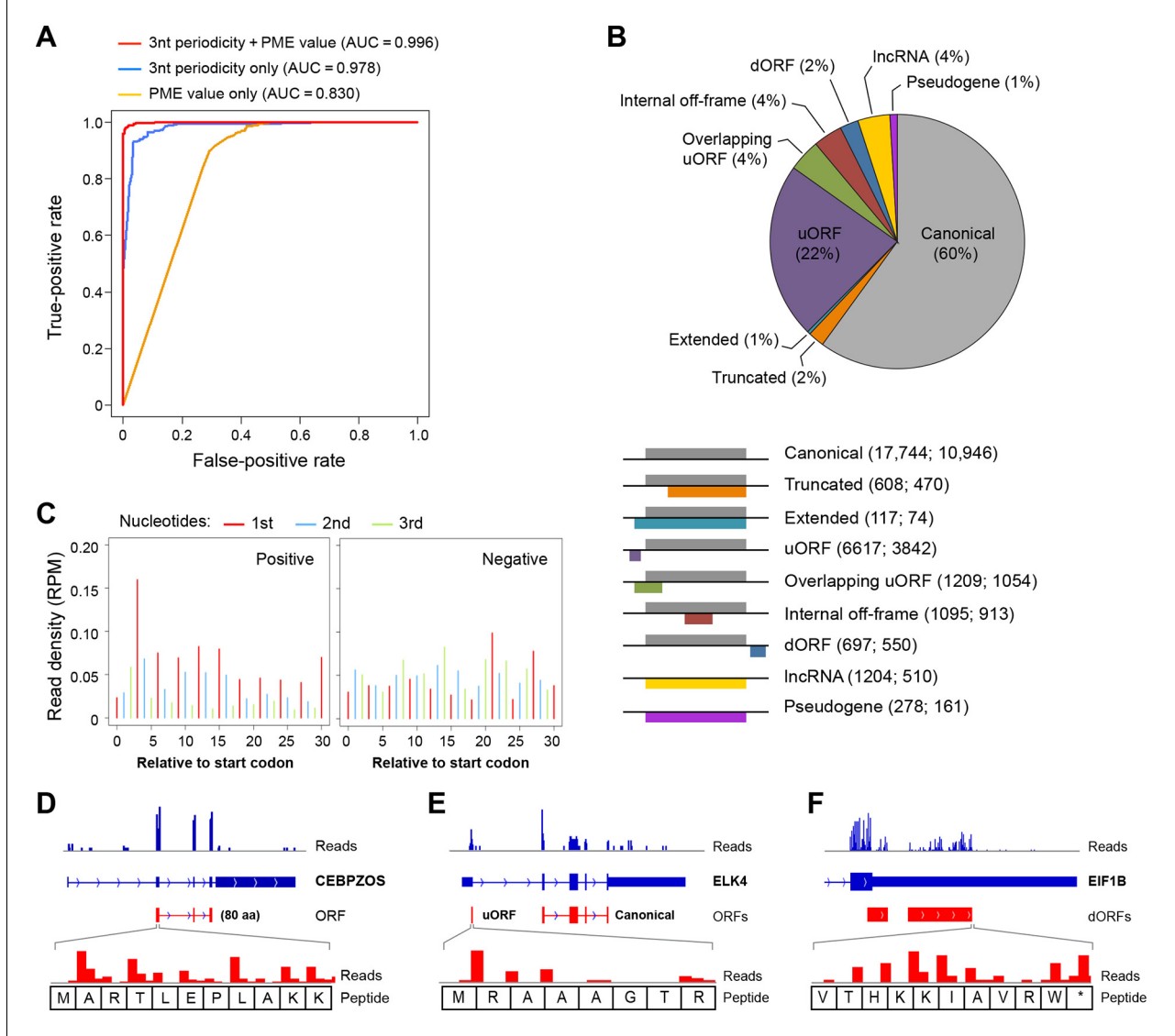

**Figure 2.** RibORF identifies translating ORFs. (**A**) Receiver-operating characteristic (ROC) curves to measure algorithm performance using different training parameters. (**B**) Types of translated ORFs identified in this study, with ORF number:gene number shown in parenthesis. (**C**) Distribution of reads upon cycloheximide treatment around start codon of predicted positive and negative lncRNA ORFs. Examples of (**D**) a translated lncRNA (**E**) an mRNA with a uORF (**F**) an mRNA with a dORFs; the 3' most exon is shown. Enlarged figures show 3-nt periodicity can be observed for each codon in *Figure 2D–F*.

The following figure supplements are available for figure 2:

**Figure supplement 1.** RibORF algorithm performance.

**Figure supplement 2.** Analysis of ribosome-associated RNA.

probabilities are well correlated in the two cancer models (R = 0.97), indicating the algorithm can be robustly applied to various cell types (*Figure 2—figure supplement 1B*).

We applied the classifier to predict translated ORFs within lncRNAs, pseudogenes, and mRNAs. Candidate ORFs showed a mixed population of 3-nt periodicity and PME values (*Figure 1C,G*). Using a stringent cutoff for the probability of prediction (0.7 with a false positive rate 0.67% and a false negative rate 2.5%; *Figure 2—figure supplement 1C*), we identified canonical ORFs in 10,946 protein-coding genes, and truncated or extended variants in 544 genes (*Figure 2B*). The canonical ORFs in almost all expressed transcripts were identified. In addition, we identified so-called uORFs in the 5'UTRs of 3842 protein-coding genes, and uORFs overlapping with coding regions (overlapping uORFs) in 1054 genes *Figure 2B*). We also identified ORFs located in 3'UTRs of 550 genes, which we term downstream ORFs (dORFs; *Figure 2B*). In general, translated uORFs and dORFs are expressed from the same transcript as the relevant canonical ORF, although in some cases these may arise from truncated transcripts. Lastly, we identified 1204 ORFs in 510 lncRNAs and 278 ORFs in 161 pseudogenes (*Figure 2B*). As expected, the predicted translated ORFs show clear 3-nt periodicity and high PME values, while the negative ones do not (*Figure 2C* and *Figure 2—figure supplement 1C-D*). Examples of lncRNA ORFs, uORFs and dORFs are shown in *Figure 2D--F*, and a full list is presented in *Supplementary file 1*. For the well expressed ORFs, we observe 3-nt periodicity for individual codons (*Figure 2D–F*).

Uniform 3-nt periodicity over an extended distance is diagnostic of *bona fide* translation. In this regard, all 7 tested RNAs encoding non-canonical translated ORFs are associated with 80S monosomes and/or polysomes (*Figure 2—figure supplement 2*). Thus, we will refer to the products of translated ORFs as 'peptides', even though direct biochemical evidence is lacking. In this regard, the peptides represent initial translation products whose stability in vivo is unknown. We suspect that many non-functional peptides will be degraded rapidly and hence difficult to detect biochemically.

## Nuclear/cytoplasmic localization is a major determinant of translation efficiency

We did not detect translation for 679 lncRNAs in breast epithelial cells even though RNA-seq analysis indicates that they are expressed at comparable levels to the 510 translated lncRNAs (p>0.05; *Figure 3A*). We hypothesized that the distinction between these two classes is that the untranslated lncRNAs would be preferentially localized in nucleus and not accessible to the translation machinery, whereas the translated lncRNAs would be preferentially localized in the cytoplasm. To test this hypothesis, we examined the cytosolic and nuclear distribution (C:N ratio) of lncRNAs, using RNA-seq data from multiple cell lines (*Djebali et al., 2012*; *ENCODE, 2012*). Indeed, untranslated lncRNAs are less likely to localize to the cytoplasm (lower C:N ratio), than translated ones (p<$10^{-70}$; *Figure 3B*). Similar results are observed for lncRNAs in a variety of cell lines (*Figure 3—figure supplement 1A–D*). Compared to canonical protein coding mRNAs, translated lncRNAs show slightly lower C:N ratios (p<$10^{-46}$; *Figure 3B*). Translated pseudogene RNAs are also more likely to be localized in the cytoplasm as compared with untranslated pseudogene RNAs (*Figure 3—figure supplement 1E–G*).

Translation efficiency of a given RNA is defined as the ratio of translated RNA (from ribosomal profiling): overall RNA (from RNA-seq). In accord with the reduced C:N ratio of translated lncRNAs as compared to mRNAs, lncRNAs also show lower translation efficiency (p<$10^{-12}$; *Figure 3C*). However, when corrected for the reduced levels of lncRNAs in the cytoplasm, it appears that the translation efficiency of cytoplasmic lncRNAs and mRNAs are nearly comparable, albeit slightly reduced. Interestingly, the translation efficiencies of mRNAs vary hundreds of fold (*Ingolia et al., 2009*) (*Figure 3D*), and these differences are strongly correlated with localization in the cytosol (*Figure 3E* and *Figure 3—figure supplement 1H–I*).

The strong relationship between nucleo-cytoplasmic location and translatability of lncRNAs provides strong independent evidence that our classifier effectively identifies translated RNAs. In addition, translation efficiency is strongly correlated with degree of cytoplasmic location, indicating that accessibility of an RNA to the translation machinery is a major determinant of how well it is translated.

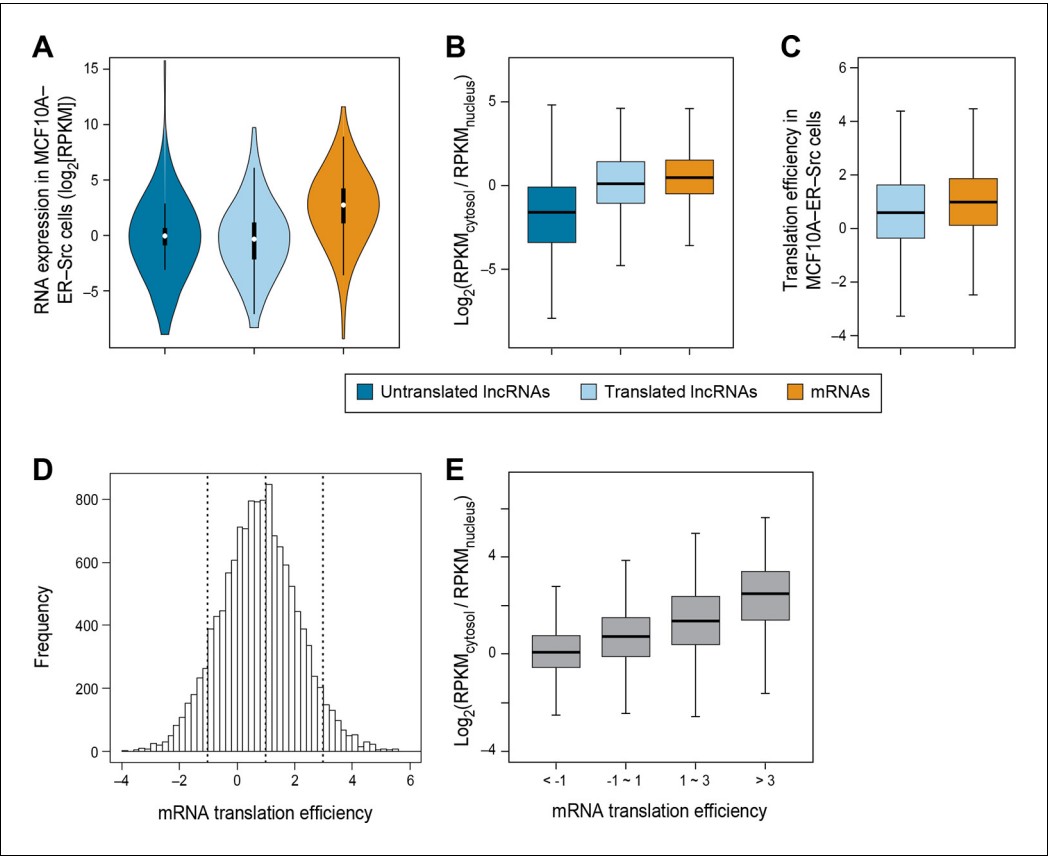

**Figure 3.** RNA subcellular localization is a major determinate of translation efficiency. (**A**) RNA expression levels of lncRNAs with or without translated ORFs and canonical mRNAs in MCF10A-ER-Src cells. (**B**) Relative subcellular location of translated and untranslated lncRNAs and canonical mRNAs. (**C**) Translation efficiency of translated lncRNAs and canonical mRNAs. (**D**) Distribution of translation efficiency of canonical mRNAs, calculated as averaged translation efficiency values in breast epithelial and fibroblast cells. (**E**) Relative subcellular locations of mRNAs grouped based on translation efficiency.

The following figure supplement is available for figure 3:

**Figure supplement 1.** RNA subcellular localization regulates translation.

## Features of lncRNA peptides

Over 40% (491 out of 1189) of expressed lncRNAs encode peptides longer than 10 aa, and 8% (98 lncRNAs) encode peptides longer than 100 aa (*Figure 4A*). The median length of all peptides translated from lncRNAs (43 aa; *Figure 4B*) is considerably longer than that of peptides generated from uORFs (17 aa). Translation of many lncRNAs yields multiple peptides from non-overlapping ORFs, and the median length of the longest peptide translated by a given lncRNA is 62 aa (*Figure 4C*). Translated lncRNAs use AUG start codons more often than uORFs (*Figure 4—figure supplement 1A,B*).

For mRNAs, the longest candidate ORFs are virtually always translated into functional proteins, but this is not the case for lncRNAs. The median length of the longest candidate ORF in a given lncRNA is 79 aa, but the longest candidate ORFs is translated only for 56% of the lncRNAs (*Figure 4—figure supplement 1C,D*). For the remaining 38% of the lncRNAs, the translated ORF was located upstream of the longest ORF. This preferential translation of ORFs located closer to the 5' ends of the lncRNAs likely reflects the strong preference of translation to initiated at the first AUG codon. The fact that the longest candidate ORF and/or its 5' proximal location is not necessarily the portion of the lncRNA that is translated indicates the value of the RibORF algorithm.

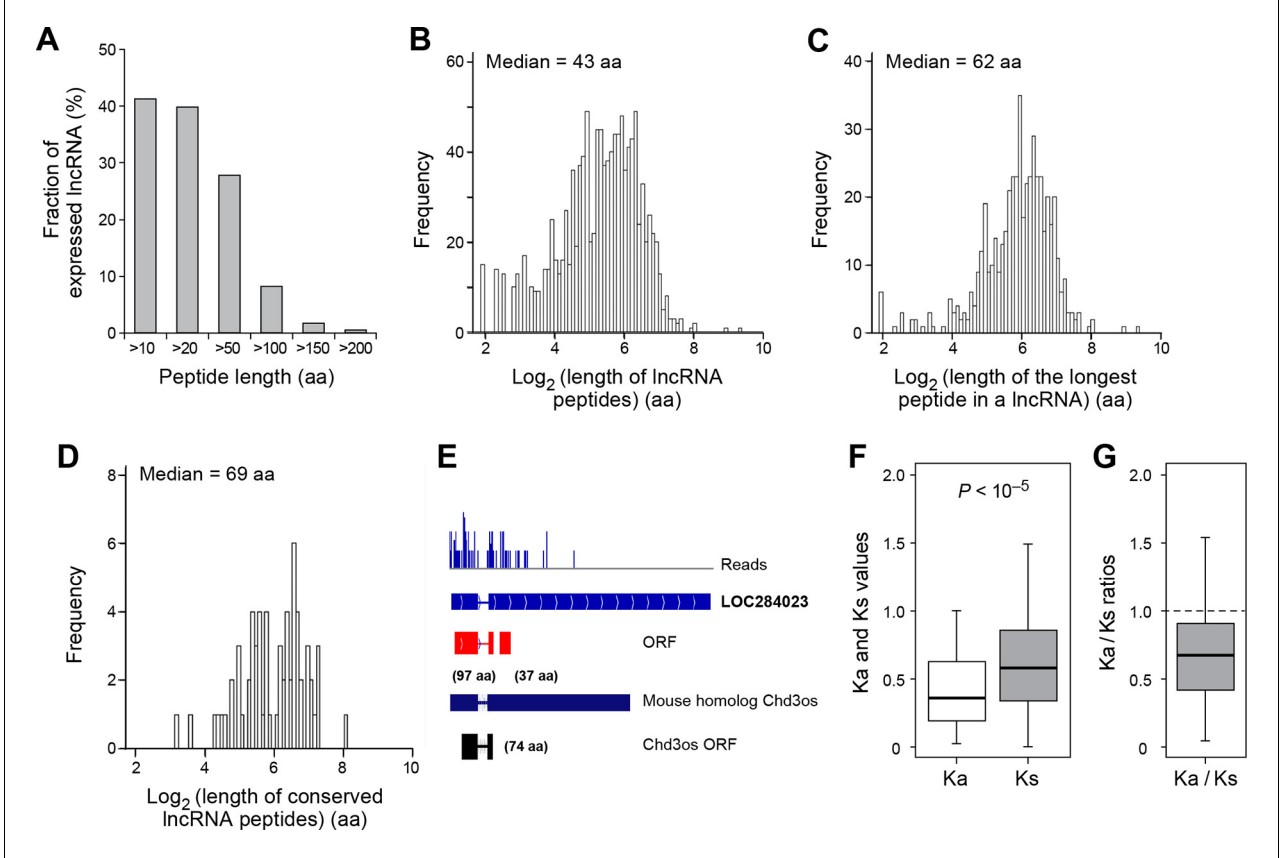

**Figure 4.** Features and conservation of lncRNA peptides. (**A**) Fraction of expressed lncRNAs that encode peptides longer than a certain length. (**B**) Peptide length encoded by lncRNAs. (**C**) Length of the longest peptide in a given lncRNAs. (**D**) Length of conserved lncRNA peptides. (**E**) LncRNA LOC284023 encodes two peptides, the upstream one being conserved in the mouse lncRNA Chd3os. (**F**) Ka and Ks values of types of conserved lncRNA peptides with Z-Test p-values shown. (**G**) Ka/Ks ratios of types of conserved lncRNA peptides.

The following figure supplements are available for figure 4:

**Figure supplement 1.** Features of lncRNA translation.

**Figure supplement 2.** Conservation of nucleotides encoding lncRNA and pseudogene peptides.

**Figure supplement 3.** Coding potential of nucleotides encoding lncRNA and pseudogene peptide.

**Figure supplement 4.** BLASTP E-values of peptide sequences encoded by homologous human and mouse ORF.

**Figure supplement 5.** BLASTP E-values of peptide sequences encoded by homologous human and mouse peptides.

**Figure supplement 6.** The Ka/Ks ratios between human translated ORFs and 50 randomly generated sequences with BLASTP alignment E-value <$10^{-4}$.

## Conservation of human lncRNA peptides in mouse

To address the functional significance of peptides translated from lncRNAs, we used four approaches to study their evolutionary conservation. First, we used PhastCon scores based on 44-vertebrate Multiz alignment (*Siepel et al., 2005*) to measure conservation of ORF nucleotide sequence among species (*Figure 4—figure supplement 2*). Second, we used the PhyloCSF score to study the protein-coding potential of ORF sequences based on 29-mammal genome alignment (*Lin et al., 2011*)(*Figure 4—figure supplement 3*). Third, we checked the conservation of human peptides in mouse transcripts at the amino acid level and defined them to be conserved if two

homologous ORFs encode peptides with a BLASTP alignment E-value $<10^{-4}$ (False Discovery Rate < 0.0005 for all types and lengths of ORFs; *Figure 4—figure supplements 4* and *5*, and *Supplementary file 2*). Fourth, for lncRNA peptides conserved between human and mouse, we computed the ratio of nonsynonymous (Ka) to synonymous (Ks) substitution rates of the homologous nucleotide sequences. The Ka/Ks ratio is a commonly used parameter to infer the direction and magnitude of natural selection on peptide sequences (*Hurst, 2002*). A ratio smaller than 1 indicates a significant number of nucleotide sequence changes that do not result in protein sequence changes, indicating that the protein is under stabilizing (negative) selection and likely to be functional. For these analyses, we excluded the 30 lncRNAs that encode peptides conserved in mouse protein-coding genes and likely to be pseudogenes mis-annotated by GENCODE (*Supplementary file 2*).

For each translated ORF, we compared its conservation level (Phastcon and PhyloCSF score) to untranslated segments that are matched for length and transcript location. Interestingly, at the nucleotide level, translated ORF sequences tend to be more conserved and have higher coding potential than the untranslated sequences (p<$10^{-4}$; *Figure 4—figure supplements 2A* and *3A*). The pattern is consistent for translated ORFs with different lengths, suggesting that some peptides might be functional. Most lncRNA peptides (92%) do not contain protein domains annotated by Pfam (*Punta et al., 2012*) (*Figure 4—figure supplement 2C*). ORF nucleotide sequences encoding short peptides (<100 aa) containing protein domains are more conserved (p<$10^{-3}$; *Figure 4—figure supplement 2D*).

93 translated lncRNAs (19% of the total) have homologous lncRNA genes in mouse. From those conserved lncRNA genes, 41 (44%) express conserved peptides, with a median length 69 aa (*Figure 4D*, *Figure 4—figure supplement 4A*, and *Supplementary file 2*). As expected, these conserved peptides have higher coding potential than non-conserved ones (*Figure 4—figure supplement 3A*). For example, the human lncRNA LOC284023 expresses a 97 aa peptide encoded by the 5' end, and a 37 aa peptide encoded downstream (*Figure 4E*). The 97 aa peptide is conserved in mouse homologous transcript Chd3os, while the 37 aa peptide is not. Interestingly, human lncRNA peptides conserved with mouse peptides encoded by lncRNAs have Ka/Ks ratios significantly lower than 1 (*Figure 4F,G*). The low Ka/Ks ratios were not due to our BLASTP E-value cutoff (*Figure 4—figure supplement 6*). 20 such lncRNAs express peptides with Ka/Ks values smaller than 0.5, and 12 have values < 0.3. Consistently, peptides with lower Ka/Ks values have higher coding potential based on PhyloCSF scores (*Figure 4—figure supplement 3A*), suggesting that they are evolutionary stabilized and are probably functionally important.

## Features and conservation of pseudogene peptides

The human genome contains 13,708 annotated pseudogenes that are derived from ancestral protein-coding genes but generally not expressed as RNAs and believed to have lost their protein-coding capability. However, out of 426 expressed pseudogenes (~3% of those annotated), 155 (36%) are translated into peptides longer than 10 aa. In addition, 81 expressed pseudogenes (19%) generate peptides longer than 100 aa (*Figure 5A*), and most (~80%) of these contain at least one protein domain (*Figure 4—figure supplement 2C*). The median length of pseudogene peptides is 70 aa (*Figure 5B*), and the median length of the longest peptide translated by a pseudogene is 102 aa (*Figure 5C*), which is 30 aa longer than lncRNA peptides.

Nucleotide sequences of translated ORFs in pseudogenes are significantly more conserved and have higher coding potential than untranslated sequences of the matching sizes and relative positions, and the pattern is consistent for translated ORFs of various sizes (p<$10^{-22}$; *Figure 4—figure supplements 2B* and *3B*). 114 pseudogene peptides (74% out of those translated) are conserved in mouse, with a median length 92 aa (*Figure 5D*, *Figure 4—figure supplements 3B* and *4B*, and *Supplementary file 2*) that is ~25% the length of the corresponding canonical proteins. For example, the mouse protein-coding gene Fam86 has a homologous protein-coding gene FAM86A in human, and also has a homologous pseudogene FAM86C2P, which is annotated as a long noncoding RNA. We found FAM86C2P is translated into a peptide with 131 aa, while mouse Fam86 protein is 336 aa (*Figure 5E*). Several internal coding exons in Fam86 are lost in FAM86C2P during evolution.

69% of conserved human pseudogene peptides are homologous to canonical ORFs in mouse mRNAs (*Figure 5F*). As a class, these conserved peptides show a Ka/Ks ratio significantly lower than 1 (*Figure 5G,H*), with 50 pseudogenes expressing peptides with Ka/Ks values lower than 0.3. This

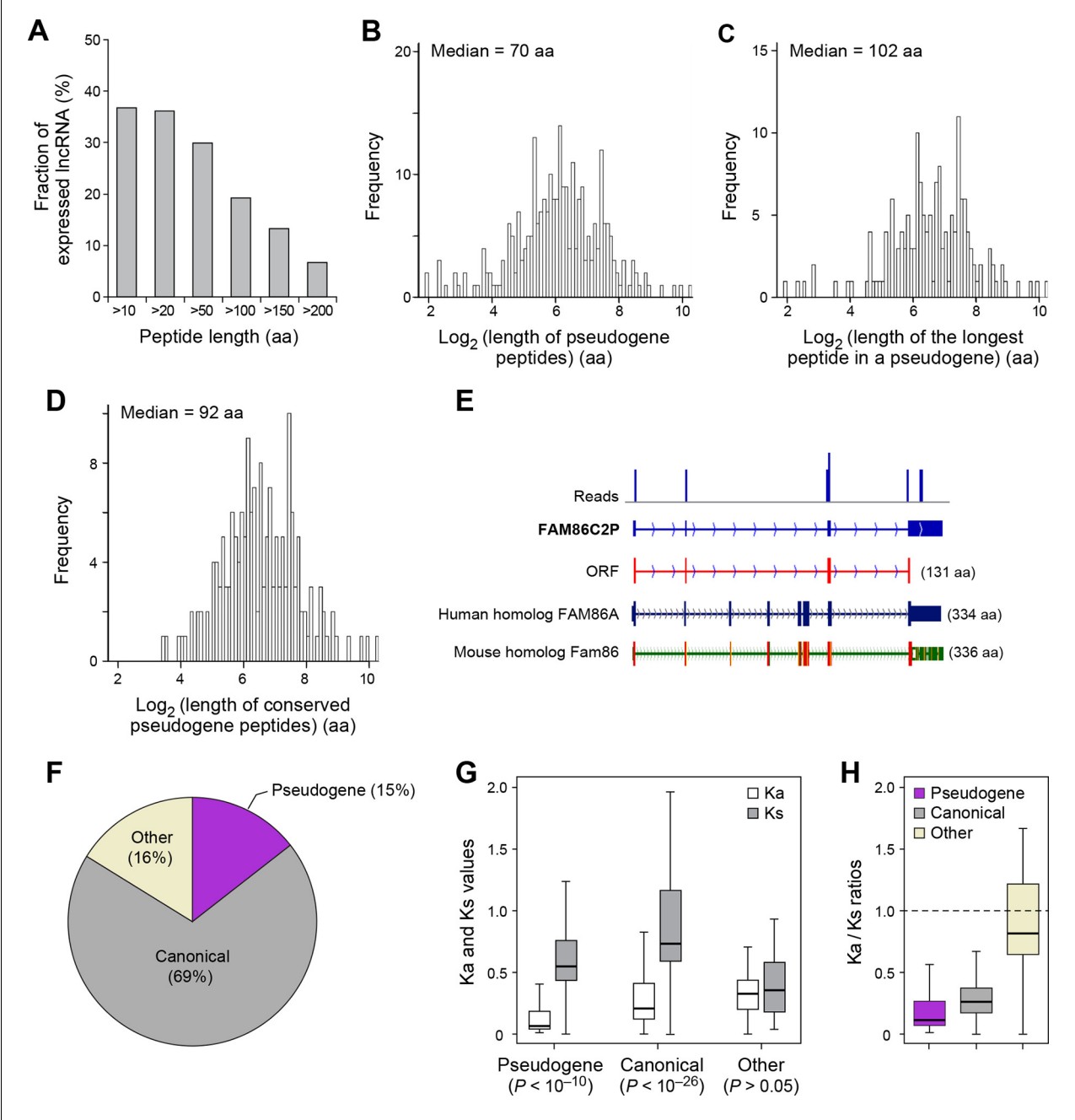

**Figure 5.** Features and conservation of pseudogene peptides. (**A**) Fraction of expressed pseudogenes that encode peptides longer than a certain length. (**B**) Peptide length encoded by pseudogenes. (**C**) Length of the longest peptides in a given pseudogenes. (**D**) Length of conserved pseudogene peptides. (**E**) Peptide in a human pseudogene FAM86C2P is conserved in the mouse protein coding gene Fam86. FAM86C2P also has a homologous human protein coding gene FAM86A. (**F**) Conserved human pseudogene peptides, grouped based on their homologous ORF types in mouse genome. (**G**) Ka and Ks values of types of conserved pseudogene peptides with Z-Test p-values shown. (**H**) Ka/Ks ratios of types of conserved pseudogene peptides.

suggests that, although some human pseudogenes are translated into shorter peptides than their mouse homologs, the peptide sequences are evolutionarily constrained, and hence may play functional roles. In addition, 15% of conserved pseudogene peptides are homologous to mouse pseudogenes, and these peptides also have Ka/Ks ratios even lower than those homologous to mouse

canonical ORFs, including 19 with Ka/Ks ratios < 0.3 (*Figure 5F–H*). Thus, pseudogenes with longer evolutionary histories are more likely to encode functional peptides. In contrast, the remaining 16% of conserved pseudogene peptides are homologous to non-canonical ORFs in mouse mRNAs, and these peptides have Ka/Ks ratios close to 1 suggesting they are nonfunctional (*Figure 5F–H*).

## Translation of uORFs and dORFs and the relationship to protein-coding sequences

The median lengths of uORFs (17 aa) and overlapping uORFs (37 aa) are shorter than those of lncRNAs and pseudogene peptides (*Figure 6A*). In general, the translation efficiency of uORFs is similar to that of canonical protein-coding sequences (*Figure 6B*), and this effect is typical for individual genes. However, in accord with previous results linking uORFs to decreased protein levels (*Calvo et al., 2009*; *Barbosa et al., 2013*), the translational efficiency of mRNA coding regions is slightly lower for genes containing uORFs (p<10$^{-34}$; *Figure 6C*), even though RNA levels of uORF-containing genes somewhat higher than genes lacking uORFs (p<10$^{-200}$; *Figure 6D*). However, the

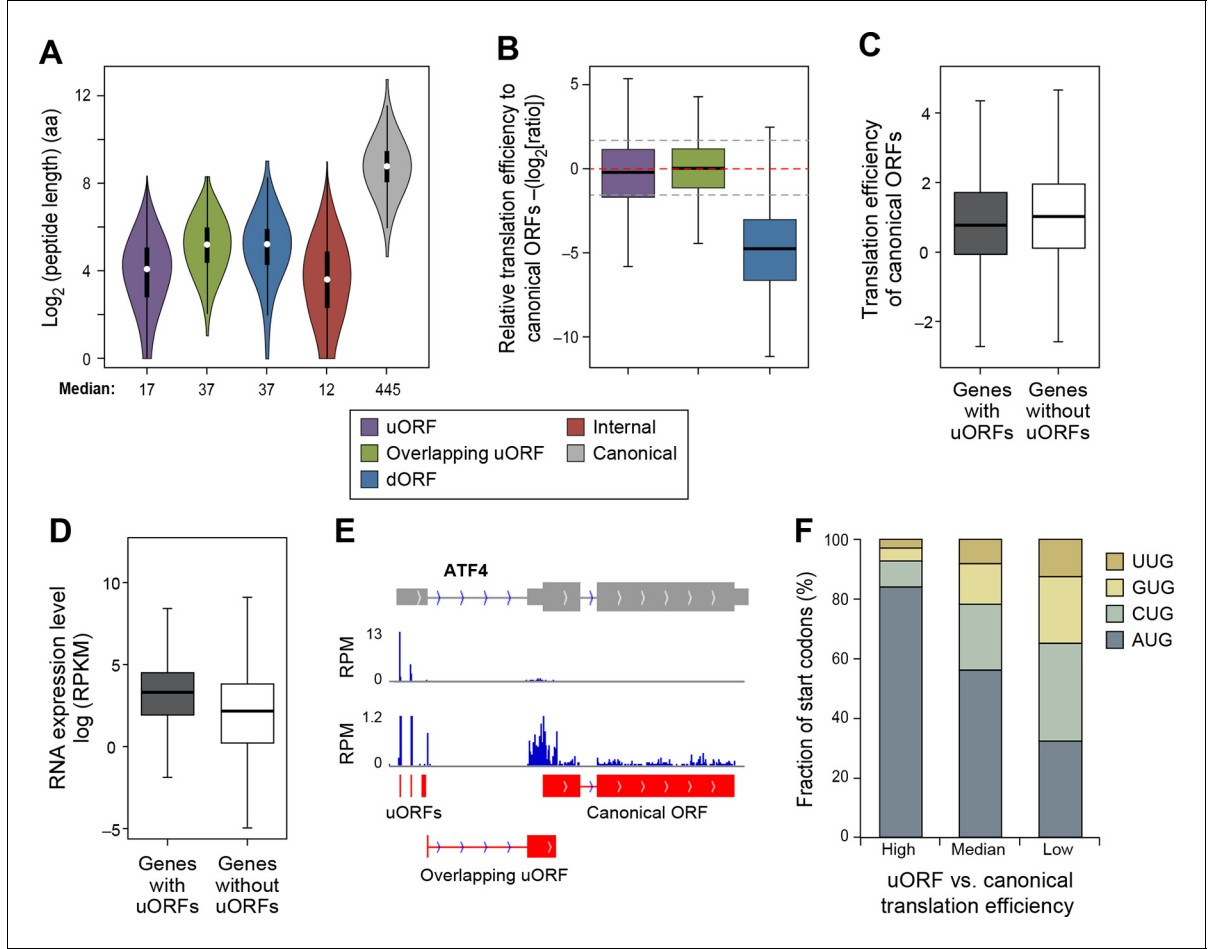

**Figure 6.** Features of ORFs encoded by protein coding genes. (**A**) Length distribution of peptides encoded by human protein coding genes. (**B**) Relative translation efficiency comparing non-canonical ORF vs. canonical ORF from the same gene. (**C**) Translation efficiency of canonical ORFs comparing genes with/without uORFs. (**D**) RNA expression level of genes with/without uORFs, measured by RNA-seq. (**E**) ATF4 encoded 3 uORFs and 1 overlapping uORF, whose translation efficiency is much higher than the canonical ORF. (**F**) Start codon types of uORFs showing differential relative expression levels to canonical ORFs. High: >three-fold higher than canonical ORFs. Low: >three-fold lower than canonical ORFs.

The following figure supplement is available for figure 6:

**Figure supplement 1.** Example genes showing high translation of uORFs.

relatively high translational efficiency of protein-coding regions in genes containing uORFs suggests that scanning ribosomes often skip the uORF to allow efficient initiation at the protein-coding ORF.

Interestingly, 1,1144 genes show >three-fold higher translational efficiency of the uORF than the corresponding protein-coding region, suggestive of translational regulation in a manner similar to Gcn4 (*Supplementary file 3*) (*Hinnebusch, 2005*). These are enriched for 'transcription regulators' (p<10$^{-8}$; Fisher's Exact Test; 237 genes are in the pathway 'regulation of transcription; GO:0045449'), particularly zinc finger transcription factors (p<10$^{-9}$; Fisher's Exact Test), and protein kinases (p<10$^{-5}$; Fisher's Exact Test; 45 genes are in the pathway 'protein kinase cascade GO:0007243'). Interestingly, many AP-1 transcription factors (ATF4, ATF5, ATF2, and JUN) have high usage of uORFs, similar to the yeast homolog Gcn4. For example, ATF4 contains 3 uORFs and 1 overlapping uORF, and the uORF expression is over 300-fold higher than the canonical ORF under normal growth conditions (*Figure 6E*). However, under stress conditions, ATF4 efficiently re-initiates translation of the canonical ORF, thereby resulting in higher protein expression (*Rutkowski and Kaufman, 2003*; *Vattem and Wek, 2004*). Many other regulatory genes (e.g. RELA, PTEN and DICER1) also show high uORF usage and suppressed translation of the canonical protein regions (*Figure 6—figure supplement 1*). The major determinant of uORF translation efficiency is its start codon as 84% of highly translated uORFs (>three-fold higher than canonical ORFs) use AUG as start codon, while only 32% of poorly translated uORFs (>three-fold lower than canonical ORFs) use AUG (*Figure 6F*).

In contrast to uORFs, the translation efficiency of dORFs is much lower (30-fold on average) than the corresponding protein-coding region, indicating a very low level of translational reinitiation after the canonical stop codon (*Figure 6B*). However, a small subset of dORFs are translated much more efficiently than the average dORF (*Supplementary file 3*).

## Conservation and possible biological function of uORF and dORF peptides

Using the analytical methods described above, we found that nucleotide sequences encoding uORFs and dORFs are more conserved than neighboring untranslated sequences, with 20% human uORF peptides, 46% of overlapping uORF peptides, and 32% of dORF peptides are conserved in mouse (*Figure 7A*, *Figure 7—figure supplement 1*, and *Supplementary file 2*). Interestingly, these peptides have Ka/Ks ratios significantly lower than 1, suggesting they may play functional roles (*Figure 7B,C*, and *Figure 7—figure supplement 2*). While uORFs clearly have an important role in inhibiting downstream expression of the canonical protein (*Morris and Geballe, 2000*; *Barbosa et al., 2013*) (*Figure 6D,E*) our results suggest that some of the encoded peptides are under stabilizing selection.

## Discussion

### RibORF, an improved method for mapping translated regions in vivo

Although ribosome-profiling experiments indicate that lncRNAs and non-canonical ORFs in mRNAs can be translated (*Ingolia et al., 2011*; *2014*; *Aspden et al., 2014*; *Bazzini et al., 2014*; *Ruiz-Orera et al., 2014*), previous methods to identify the translated products have been problematic. First, with one exception (*Bazzini et al., 2014*), they did not use 3-nt periodicity to identify translated ORFs, but rather relied on the longest ORF length and/or maximum read density, which does not provide clear evidence for in-frame translation. Second, with one exception (*Ingolia et al., 2014*), they did not filter the many reads that arise from non-ribosomal complexes and hence are irrelevant to identifying translated proteins. Third, we account for the variable distances between the 5' end of the sequenced fragment and the ribosome A-site that arises due to imperfect RNase trimming of RPFs, and include RPFs with variable lengths into analyses for maximum sequencing read usage and codon coverage. Alignment of A-sites is critical for observing optimal 3-nt periodicity that characterizes translated regions. With this step, we can observe clear 3-nt periodicity for each codon in well expressed translated ORFs. Fourth, with approaches using harringtonine or lactimidomycin treatment to block translational elongation and hence map the translation initiation site (*Ingolia et al., 2011*; *Lee et al., 2012*), additional experiments are required, read peaks are often not precisely located at the start codons, and many genes do not show efficient ribosome pausing

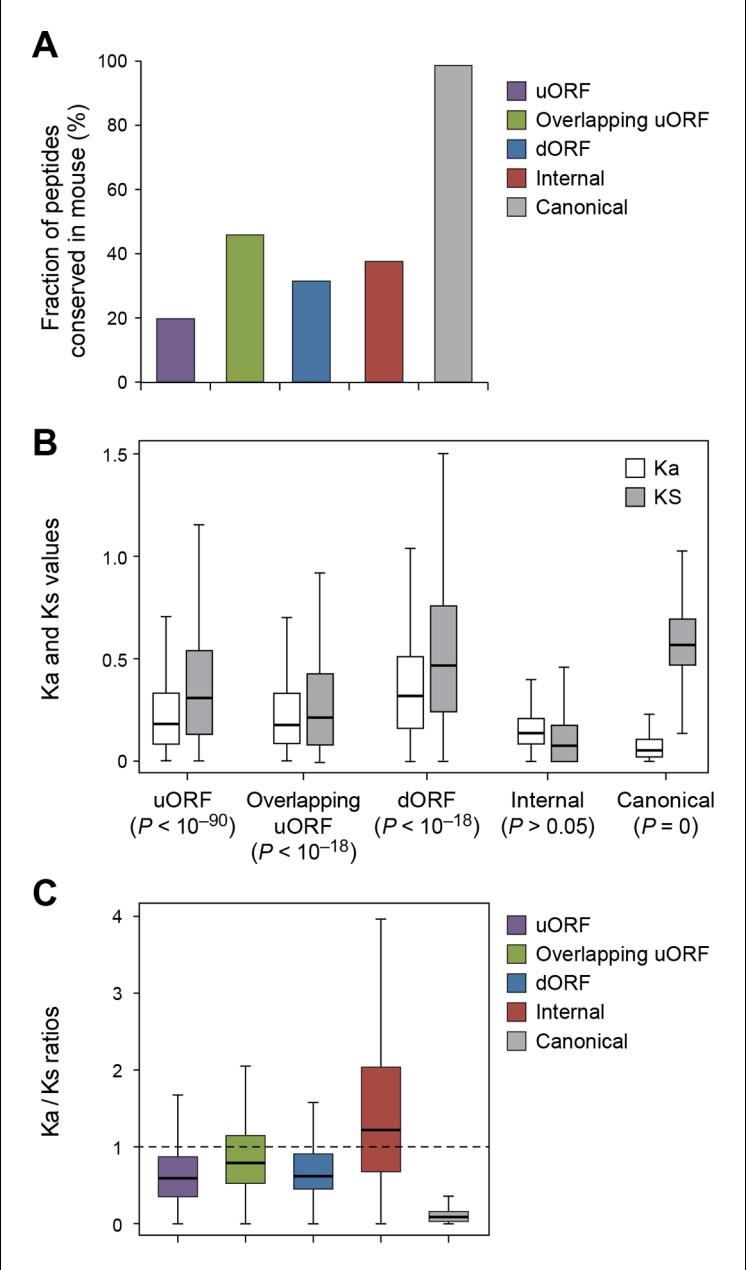

**Figure 7.** Conservation of non-canonical peptides encoded by mRNAs. (**A**) Fraction of human mRNA peptides conserved in mouse. (**B**) Ka and Ks values of conserved mRNA peptides with Z-Test p-values shown. (**C**) Ka/Ks ratios of conserved mRNA peptides.

The following figure supplements are available for figure 7:

**Figure supplement 1.** Conservation of nucleotides encoding uORF and dORF peptides.

**Figure supplement 2.** Examples of conserved uORF peptides.

at start codons. In addition, the ribosome profiling datasets involving mammalian cells that have been analyzed to date were generated by a polyA tailing procedure that causes inaccuracies in determining the true 5′ end of the RPF.

The RibORF algorithm combines ribosome A-site alignment, 3-nt periodicity and uniformity across codons (PME approach) to define regions of active translation. Using this approach, we identify a few thousand non-canonical peptides translated from lncRNAs, 5' UTRs, and 3'UTRs, a dramatic increase over the several hundred previously reported (*Bazzini et al., 2014*). In addition, we show that, although the vast majority of pseudogenes are not transcribed, 36% of expressed pseudogenes are translated into peptides. We believe that the RibORF approach represents a significant improvement over current methods, and it should be generally applicable.

## Many lncRNAs are translated, but most encoded peptides are likely to be nonfunctional

Approximately 40% of lncRNAs are translated into peptides >10 aa in length, with a median length of 43 aa. The distinction between translated and untranslated lncRNAs is strongly correlated with whether they are or not in the cytoplasm. Furthermore, the translation efficiency of lncRNAs is comparable to that of mRNAs, indicating that simple access of an RNA to the translation machinery in the cytoplasm is a major determinant of how well it is translated. In this regard, lncRNAs are transcribed by RNA polymerase II, capped, and polyadenylated, and hence are largely indistinguishable from mRNAs with respect to translation. It is unclear why some lncRNAs are predominantly nuclear, whereas others are predominantly cytoplasmic, but it seems unlikely that this is simply a matter of chance. It will be interesting to study whether nucleo-cytoplasmic localization of lncRNAs can be regulated during biological processes. However, the translatability of lncRNAs *per se* does not indicate whether the peptide is biologically important or even sufficiently stable to be detected.

The observation that cytoplasmic lncRNAs are translated suggests that many of the resulting peptides are not themselves biologically functional. In this regard, the majority of lncRNA-encoded peptides are not conserved in mouse or other species. This lack of conservation does not exclude the possibility that these peptides are biologically meaningful, but it seems likely that most lncRNA peptides are not. If so, biological function may be mediated by the lncRNAs themselves or by the act of Pol II transcription, which alters chromatin structure to affect processes such as Pol III transcription (*Moqtaderi et al., 2010*; *Oler and Cairns, 2010*) and V(D)J recombination (*Matthews et al., 2007*). However, the possibility remains that some or many lncRNAs represent transcriptional noise (*Struhl, 2007*).

## A subset of peptides encoded by lncRNAs, pseudogenes, 5'UTRs, and 3'UTRs are likely to be biologically functional

Although many, and perhaps most, of the various forms of non-canonical peptides are nonfunctional, our results strongly suggest that a minority of them is biologically functional. Some of these non-canonical peptides are conserved in mouse and as a class, these peptides have synonymous and nonsynonymous amino acid substitution rates indicating that are under stabilizing selection and strongly suggesting that they perform biological functions. Our results do not indicate that all of the conserved peptides are biologically functional, nor do they identify specific peptides as being functional. Most likely, functional peptides are those with the lowest Ka/Ks ratios, but these ratios need to be corrected for the number of substitutions analyzed for a given conserved region and the probability that they occur by chance. Nevertheless, a few functional lncRNA peptides have been described in other species (*Galindo et al., 2007*; *Kondo et al., 2010*; *Magny et al., 2013*; *Pauli et al., 2014*), and our results strongly suggest that a significant minority of non-canonical peptides have biological functions. Some previous analyses of lncRNAs tried to eliminate lncRNAs producing potentially functional peptides and removed those with long ORF length, high conservation and protein domains using various cutoffs (*Cabili et al., 2011*; *Harrow et al., 2012*). Here we found many lncRNAs after the filtering are translated, and conserved lncRNA and pseudogene peptides have median length 69 aa and 92 aa, respectively, which is shorter than the typical cutoff 100 aa. Our results indicate that ribosome profiling provides significant values to effectively identify translated RNAs in an unbiased manner, and reveal potential functional short peptides.

## The semantics of non-coding RNAs

By definition, non-coding RNAs are not translated into protein. However, until the advent of ribosome profiling that directly identifies translated regions of RNAs in an unbiased fashion, non-coding

RNAs were defined computationally as lacking ORFs of significant length. Secondary bioinformatic considerations such as codon usage, evolutionary conservation, and protein domain have also been used as part of the definition of non-coding RNAs (*Cabili et al., 2011*; *Harrow et al., 2012*).

Here we show that ~40% of so-called lncRNAs and pseudogene RNAs are translated in vivo, and hence are not truly non-coding RNAs. Of course, the translation of these RNAs provides no information on whether the resulting peptides are stable/detectable or biologically meaningful. Our results suggest that many, and perhaps nearly all, peptides generated from lncRNAs and pseudogene RNAs arise from the invariable translation of the cytoplasmic RNAs, thereby fortuitously generating peptides of no biological consequence. In such cases, any biological function of these RNAs would depend on the RNA product itself. However, the evolutionary conservation and low Ka/Ks ratios of some peptides generated by lncRNAs and pseudogene RNAs are suggestive that these peptides confer some biological function. An RNAs that generates a functional peptide may also have a biological function as an RNA molecule.

Thus, non-coding RNAs can be divided into 3 classes, namely 1) true non-coding RNAs that are not translated, 2) RNAs that are translated into functionally irrelevant peptides, and 3) RNAs that are translated into non-conventional proteins that confer biological function. Furthermore, as the nucleo-cytoplasmic location of RNAs might be regulated by cell-type or environmental conditions, some RNAs that appear to be truly non-coding in our experiments might be translated and give rise to functional peptides in other circumstances.

## Materials and methods

### Cell culture

All cultures were performed at 37°C under 5% $CO_2$. BJ fibroblast cell lines (EH, EL and ELR) were cultured on Knockout DMEM (Thermo Fisher Scientific, Waltham, MA) with 10% FBS, medium 199, glutamine and penicillin-streptomycin (*Hahn et al., 1999*). The breast epithelial cell line (MCF10A-ER-Src) was grown in DMEM/F12 with 5% charcoal-stripped fetal bovine serum (Thermo Fisher Scientific, Waltham, MA) and supplements (*Iliopoulos et al., 2009*).

### Ribosome-profiling and RNA-seq library preparation

Cells were seeded at $1 \times 10^6$ cells per 10-cm culture dish and cultured overnight. MCF10A-ER-Src cells were treated by 1 μM 4-hydroxy-tamoxifen for various time points (1, 4, and 24 hr) to induce transformation. Cells were pretreated with cycloheximide (100 μg/ml; Sigma-Aldrich, St. Louis, MO) for 90 s or harringtonine (2 μg/ml; Santa Cruz, Santa Cruz, CA) for 5 min, and detergent lysis was then performed with flash-freezing in liquid nitrogen. For ribosome profiling, DNase I-treated lysates were then treated with RNase I, and ribosome-protected fragments were purified for Illumina TruSeq library construction as previously described (*Ingolia et al., 2012*). For RNA-seq, total RNA was purified from DNase-treated lysates, and ribosomal RNA was depleted with RiboMinus Eukaryote Kit (Thermo Fisher Scientific, Waltham, MA). RNA-seq libraries were prepared with a tagging-based workflow (*Pease and Kinross, 2013*). In brief, rRNA-depleted RNA was fragmented at 85°C for 5 min, followed by cDNA synthesis, terminal tagging and PCR amplification with ScriptSeq v2 RNA-Seq Library Preparation Kit (Epicentre, Madison, WI). Ribosome profiling and RNA-seq libraries were sequenced with Illumina HiSeq 2500.

### Ribosome profiling and RNA-seq analyses

We trimmed 3′ adapters from sequencing reads and then aligned the trimmed reads to human rRNA sequences and removed reads mapping to rRNAs (5S, 5.8S, 18S, and 28S). We then aligned remaining reads to the union of human reference transcript sequences: defined RefSeq; GENCODE lncRNAs; human body Map lncRNAs. The unmapped reads were then aligned to human reference genome sequence (hg19) using Tophat with default parameters (*Trapnell et al., 2009*).

RNA-seq reads were mapped using the same steps as ribosome profiling reads. For analysis of subcellular location, RPKM values were calculated from published RNA-seq data from nuclear and cytosolic fractions of MCF7 cells (*Djebali et al., 2012*; *ENCODE, 2012*). We required a transcript should have over 50 total RNA-seq reads for the calculation.

## Polysome analysis

MCF10A-ER-Src cells pretreated with 100 µg/ml cycloheximide for 90 s at 37°C were resuspended in 0.7 ml polyribosome lysis buffer [50 mM MOPS-NaOH at pH 7.4, 150 mM NaCl, 15 mM MgCl2, 0.5% Triton X-100, 100 mg/ml cycloheximide, 7 µl protease inhibitor cocktail (Cell Signaling Technology, Danvers, MA) and 3.5 µl SUPERase·In (Ambion, Thermo Fisher Scientific, Waltham, MA)], passed once through a 26-G needle, and incubated at 4°C for 15 min with gentle rotation. Upon centrifugation, the cleared cell lysate was loaded onto a 1050% continuous sucrose gradient and centrifuged at 36,000 rpm for 165 min at 2°C with SW41-Ti rotor (Beckman, Brea, CA). Fractions were assayed for RNA (absorbance at 260 nM) to determine the locations of the 40S and 60S subunits, 80S monoribosomes, and polyribosomes. RNA purified from these fractions was used to generate cDNA using a 1:1 combination of Oligo(dT)$_{20}$ and random hexamer and AffinityScript reverse transcriptase (Agilent, Santa Clara, CA). The ribosome-associated amount of indicated RNA from each fraction was calculated by normalizing first to the 18S rRNA amount from that fraction and second to the indicated RNA amount from unfractionated sample loaded onto sucrose gradient.

## Translation efficiency

The translation efficiency of an ORF is calculated as the log2 ratio of the ribosome profiling RPKM value: RNA-seq RPKM value. We required the ORFs to have over 10 RNA-seq and ribosome profiling reads to permit a more accurate calculation, and we excluded ORF regions overlapping with other types of ORFS.

## Transcript annotations

Protein coding genes were defined by RefSeq database. Short noncoding RNAs were defined by RefSeq database as having length < 200 nt. Pseudogenes were defined by GENCODE and to not overlap with protein-coding genes. lncRNAs were defined by a union set of RefSeq, GENCODE or Human Body Map lncRNAs (*Cabili et al., 2011*; *Harrow et al., 2012*). We required a lncRNA to have introns or a length greater than 500 nts and that it does not overlap with any protein-coding gene or pseudogene in the same strand.

## Expressed non-coding RNAs and candidate ORFs

An expressed lncRNA was defined as transcripts encoding peptides or showing significant RNA expression estimated from RNA-seq (Cutoffs Benjamini-Hochberg corrected Poisson Test p<10$^{-3}$ and >10 reads). For all types of transcripts, we identified all possible ORFs with a start codon AUG or close variants (C/U/G)UG and a stop codon. As the predicted translation probabilities are well correlated in the two cancer models (*Figure 2—figure supplement 1B*), we combined ribosome-profiling reads in the breast epithelial and fibroblast cells to identify translated ORFs. We required expressed ORFs to have RPKM > 1 in at least one cell line model and over 10 reads.

## Percentage of Maximum Entropy (PME) values to measure uniform read distribution across codons

For each ORF, we define the total read number as $N$, and the encoded peptide length as $L$. We divide the ORF into smaller regions based on $N$ and $L$ in the following way. If $N > L$, we define a region length as 1 codon. Otherwise, a region length is defined as floor($L/N$). For each region $i$ in an ORF, we calculated the fraction of reads in the region: $P(X_i) = N_i/N$, where $N_i$ represents number of reads in region $i$. We then calculate the PME value measuring the $H(X) = \sum_{i=1}^{n}(P(X_i) * log_2 P(X_i))$ uniformity of read distribution across regions as $PME = H(X)/max(H)$, where $max(H)$ is the entropy value assuming the reads are perfectly evenly distributed across codons in an ORF.

## RibORF, a support vector machine classifier for identifying translated ORFs

Read genomic locations were adjusted based on offset distance between 5' end of fragment and A-site, based on parameters shown in *Figure 1—figure supplement 1B*. The adjusted read locations were used for ORF identification, expression level calculation and visualization. For the model training, we used as a positive set canonical ORFs from coding genes, and as a negative set off-frame ORFs in protein coding regions (with start codon AUG and stop codons) and candidate ORFs in

short noncoding RNAs. We randomly picked 600 positive examples and 300 negative examples for training, and another 600 positive examples and 300 negative examples for testing. We included two features in the model, including ribosome footprinting 3-nt periodicity calculated as fraction of reads at 1$^{st}$ and 2$^{nd}$ nucleotides of codons in an ORF, and uniformity of read distribution measured by *PME* values described above. We used Support Vector Machine (R package 'e1071') to build the classifier, with five-fold cross-validation and radial basis kernel. In some cases, we can identify overlapped positive ORFs for one transcript, with the same stop codon but multiple start codons. For these cases, we first picked AUG as start codons if present. We then chose 5' most start codon as the representative one. But if there is no read between the picked one and the next downstream candidate, we chose the next one as the representative start codon.

We used the receiver operating characteristic (ROC) curve to evaluate the performance of the RibORF classifier. The ROC curve is created by plotting the true positive rate against the false positive rate at various predicted p-value cutoffs from 0 to 1. The Area Under the ROC Curve [AUC] value closer to 1 represents better performance of the classifier. As in *Figure 2A*, we used different training parameters to build the classifier, and the AUC values measuring classifier performances were plotted.

## Nucleotide sequence conservation and protein-coding potential of translated ORFs

We examined whether translated non-canonical ORFs are more conserved and have higher coding potential than untranslated sequences in the same RNAs using PhastCon scores based on multiz alignment of 46 vertebrates (*Siepel et al., 2005*) and PhyloCSF scores based on 29-mammal alignment (*Lin et al, 2011*), respectively. The PhastCon conservation level and PhyloCSF coding potential of nucleotides in a region were calculated as the average scores across nucleotides. As in *Figure 4— figure supplements 2* and *3*, for each translated ORF in lncRNAs and pseudogenes, we randomly picked 50 untranslated segments with the same length. As translated ORFs tend to be located in 5' end of transcripts, the untranslated segments located in the 5' end are twice more likely to be picked than the 3' end ones. However, as we did not observe 5' end of lncRNAs and pseudogenes are significantly more conserved than 3' end in untranslated regions, the patterns should be consistent if we do not consider the location bias. As in *Figure 7—figure supplement 1*, for translated uORFs and dORFs, we compare their conservation and coding potential levels with their neighboring untranslated regions. If the translated ORF length is *L*, the neighboring untranslated regions were defined as *L/2* region upstream the ORF and *L/2* region downstream. We excluded the translated ORFs which are located within *L/2* regions of canonical ORFs.

## Conservation of human non-canonical peptides in mouse

We used Liftover (*Karolchik et al., 2014*) to identify orthologous genomic locations of human lncRNA ORFs in mouse, and obtained possible ORFs flanking these regions, considering all coding and noncoding transcripts in mouse genome defined by refSeq and GENCODE (*Harrow et al., 2012*). Then we used BLASTP (*Johnson et al., 2008*) to compare the similarity between human and mouse ORF peptide sequences. To obtain the expected distribution of BLASTP E-values between non-conserved peptide sequences (*Figure 4—figure supplements 4* and *5*), we randomized the nucleotide sequence of each human translated ORF for 50 times and use the BLASTP to compare the human ORF peptide sequence and the randomized the sequence. We consider a human ORF to be conserved in mouse if the two ORFs have a BLASTP alignment E-value $<10^{-4}$. Using this cutoff, the False Discovery Rate (FDR) is <0.0005 for all types and lengths of non-canonical ORFs (*Figure 4— figure supplements 4* and *5*).

## Nonsynonymous and synonymous substitutions (Ka/Ks ratio) in non-canonical peptides conserved in human and mouse

The entire nucleotide and encoded peptide sequences of non-canonical peptides conserved between human and mouse were analyzed by KaKs calculator software to examine nonsynonymous (Ka) and synonymous (Ks) substitutions and the resulting and Ka/Ks values, using the approximate method 'NG' (*Wang et al., 2010*). As a control to exclude the possibility that low Ka/Ks ratios are an artifact of the our cutoff BLASTP E-value $<10^{-4}$, we calculated the Ka/Ks ratios of a given human

peptide with 50 randomly generated sequences of the same length as the homologous mouse ORF, and with BLASTP alignment E-value $<10^{-4}$.

## Protein domain annotation

We input the peptide sequences encoded by translated ORFs to the Pfam web server (http://pfam. xfam.org/search#tabview=tab1). We included both Pfam-A and Pfam-B in the analyses, and used the default cutoff E-value <1.

## Gene ontology analyses

Gene ontology analyses were done using DAVID database (*Huang et al., 2009*).

## Statistical analyses

Unless otherwise stated, p-values were calculated by the Wilcoxon Rank Sum Test.

RibORF pipeline is available at http://www.broadinstitute.org/~zheji/software/RibORF.html

## Acknowledgements

We thank Moran Cabili, Schraga Schwartz, Nir Yosef, and Marko Jovanovic for helpful discussions. This work was supported by grants to KS from the National Institutes of Health (CA 107486).

## Additional information

### Competing interests

KS: Reviewing editor, *eLife*. The other authors declare that no competing interests exist.

### Funding

| Funder | Grant reference number | Author |
|---|---|---|
| NIH Office of the Director | CA 107486 | Zhe Ji<br>Ruisheng Song<br>Kevin Struhl |
| Howard Hughes Medical Institute | | Aviv Regev |

The funders had no role in study design, data collection and interpretation, or the decision to submit the work for publication.

### Author contributions

ZJ, AR, KS, Conception and design, Analysis and interpretation of data, Drafting or revising the article; RS, Conception and design, Acquisition of data, Drafting or revising the article

### Author ORCIDs

Ruisheng Song, http://orcid.org/0000-0002-6856-4721

## Additional files

**Supplementary files**

• Supplementary file 1. Identified non-canonical human translated ORFs.

• Supplementary file 2. Human non-canonical peptides conserved in mouse.

• Supplementary file 3. uORF and dORFs with high translational efficiency (>three-fold higher than canonical ORFs).

**Major datasets**

The following datasets were generated:

| Author(s) | Year | Dataset title | Dataset URL | Database, license, and accessibility information |
|---|---|---|---|---|
| Ji Z, Song R, Regev A, Struhl K | 2015 | Ribosome profiling and RNA sequencing of MCF10A-ER-Src and fibroblast cell transformation | http://www.ncbi.nlm.nih.gov/geo/query/acc.cgi?acc=GSE65885 | Publicly available at the NCBI Gene Expression Omnibus (Accession no: GSE65885). |

**Reporting standards:** Standard used to collect data: Standard GEO.

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
