## [Decision Letter]

Thank you for submitting your work entitled "Many lncRNAs, 5'UTRs, and pseudogenes are translated and some are likely to express functional proteins" for peer review at *eLife*. Your submission has been evaluated by James Manley (Senior editor), a Reviewing editor, and two reviewers.

The reviewers have discussed the reviews with one another and the Reviewing editor has drafted this decision to help you prepare a revised submission.

Although the reviewers agree that you have addressed important aspects of the use of ribosome profiling to identify the full spectrum of the translatome, the major deficiency of this work is the validation of the data. Many reports have already documented ribosome footprinting of unexpected RNA species, but the next advance must be the demonstration that productive translation has indeed taken place. You state: "Future work including the affinity purification of 80S ribosomes and mass-spectrometry of small peptides will provide more accurate determination of the translational status of individual transcripts." However, ribosome affinity purification has been already applied (Ingolia et al., 2014; Zhou P et al., PNAS 2013) to show that the majority of noncoding RNAs, including most long intergenic noncoding RNAs, are ribosome-bound to the same extent as coding transcripts. You must by now have validation data to add to the paper.

Ribosome profiling is a powerful technology that can be applied to identify ribosome protected positions in a genome wide unbiased fashion. You revisit one of the less explored aspects of this technique, namely to what extent ribosome profiling data can be used to identify true translation events in e.g. 5' UTRs and long non-coding RNAs. The core issue is that it is unknown to which extent non-ribosome related factors, scanning ribosomes etc. can also result in "ribosome-protected" fragments and to what extent such fragments can occur in a random fashion. You weigh in on this with a new, complementary approach to defining true translation in ribosome profiling data. You develop two tests, for codon periodicity and uniform coverage (as opposed to a single, high-abundance fragment). These approaches are different than the scoring metrics previously used by the Giraldez and Guttman groups, and this new approach is well validated here. You show that lncRNA translation tends to occur on transcripts with cytoplasmic (as opposed to nuclear) localization, which is a clear prediction of any model of lncRNA translation but has not previously been tested. You also develop several lines of evidence supporting protein-level conservation constraining a subset of translated lncRNA regions. Forty-one of these are conserved in mice, and represent candidate genes encoding tiny proteins. You argue for the translation of many pseudogenes, including continued selection on the protein-coding potential of these sequences.

The major comments that the reviewers made follow. The first, regarding validation, will require additional data, while it may be possible to address the others through modifications to the text.

1) What the field is in strong need of is a study where suggested translation events are validated at a large scale with an alternative approach than ribosome profiling. This could be mass spectrometry (there are some new approaches that identify and quantify ongoing protein synthesis events) and/or association with polysomes. Such a validation would allow for benchmarking the analysis approaches that are proposed. We believe that a reasonable number of validated targets would be 7-8, for example, from the forty-one that are conserved in mice, and represent candidate genes encoding tiny proteins.

2) Gerashchenko and Gladyshev,(2014 NAR) described a strong bias in ribosome profiling studies because of the use of cycloheximide, in particular affecting uORFs (but likely also long non-coding RNAs). It is surprising that the protocol used by the authors is not discussed in this context and it does seem possible that artifacts such as those described could indeed be a factor in the present study as well. It is unclear how the analysis approach described would deal with such artifacts.

3) Pseudogenes retain substantial nucleotide-level identity with their protein-coding ancestors. Short ribosome footprints are particularly prone to mis-mapping, and the authors don't provide details on their handling of multi-mapping reads &c. They should exclude the possibility that apparent translation of pseudogenes is a result of ribosome footprints on conventional protein-coding ancestors.

4) Can the authors comment on why transcripts with ORFs of >100 (and in some cases >200) amino acids are nonetheless classified as lncRNAs?

5) The uORF-mediated regulation of ATF4 in particular is well-studied by the Harding lab among others, and the authors should cite this literature in discussing these uORFs.

6) The authors say that, "By definition, noncoding RNAs should not be translated into protein" (in the Introduction) and "By definition, non-coding RNAs are not translated into protein" (in the Discussion). In fact, this is begging the question to some extent – there may be RNAs, e.g., that function as microRNA sponges in the cytosol but have a translated uORF whose translation is important only to avoid other translation that would interfere with this non-coding function (e.g. Ulitsky & Bartel).

7) It is surprising that about 35% of all reads do not originate from the expected periodic position (Figure 1). This suggests that there is substantial randomness in the methodology, which would be expected to contribute to the stochastic characteristics of the data.

8) The approach presented relies on the 3 nt periodicity and random distribution (measured by entropy) of the reads across the ORF. From Figure 2 it appears that it is mainly the 3 nt periodicity that is driving the classification. Thus, one critical issue is to what extent such patterns can occur by chance under the multiple testing situations that are assessed. It is also unclear why we should assume that this situation cannot be the result of factors other than ribosomes or simply occur by chance when the authors states that "It is inconceivable that uniform 3nt period…".

9) The conservation analyses are in general relatively modest and it is hard to interpret whether this is a result of that there is a large fraction of false positive translation events and that the true ones are indeed conserved or whether there is an abundance of "unusual" translation events that are not conserved. Indeed many "peptides" are very short which would suggest a larger risk for false positive 3nt periodicity and uniform distribution of reads, especially for lowly expressed genes (it is not clear if there is bias for detecting more genes with low rpf counts as truly translated).

[Editors' note: further revisions were requested prior to acceptance, as described below.]

Thank you for resubmitting your work entitled "Many lncRNAs, 5'UTRs, and pseudogenes are translated and some are likely to express functional proteins" for further consideration at *eLife*. Your revised article has been favorably evaluated by James Manley (Senior editor), and the Reviewing editor and two reviewers of the original paper. The manuscript has been improved but there are some remaining issues that need to be addressed before acceptance, as outlined below. Please note especially that the reviewers commented on the use of the words "inconceivable" and "invariable," which are not appropriate and can be misleading.

Reviewer #1:

The revised manuscript addresses most of my major concerns from the original submission. In particular, it does seem that mis-mapping cannot explain potential pseudo-gene translation, and I am satisfied that the sequencing data largely reflect 80S ribosome occupancy.

I have two substantial concerns with the interpretation of the data:

1) The authors say that cytosolic lncRNAs are translated "invariably" in three places including in the Abstract. This word is too strong even for the set of lncRNAs present in this sample, and more broadly, the only invariable thing in biology is the presence of surprising exceptions.

2) The authors don't seem to consider transcript isoform variation in their interpretation of uORFs and dORFs. In yeast, translated uORFs sometimes occur on small, independent transcripts (Arribere & Gilbert 2013, Pelechano & Steinmetz 2013) and so the lack of uORF-mediated repression may reflect the fact that the uORF is not translated from the same transcript as the CDS. Likewise, the highly translated dORFs may reflect translation of extensively 5'-truncated RNAs.

Reviewer #2:

Regarding the comment on "it is inconceivable the uniform 3-nt periodicity over an extended distance can result in anything other than bona fide translation". This was mainly a concern regarding scientific style. For very few findings in science, if any, should alternative explanations be inconceivable. As discussed, 3-nt periodicity is very strong evidence for translation but as suggested by the authors it could also occur via other events and thus is not inconceivable at the single ORF level. This may not only include biological aspects but also the stochastic nature of data which will suggest 3-nt periodicity with some false positive rate. The stochastic nature of the data was the main concern of this reviewer.

1) The authors seem to agree that all the reads are not expected to be derived from RNA fragments that are protected by ribosomes. The issue that I have is that there is no background model for how this relatively large proportion of the reads would stochastically result in 3nt periodicity.

In this context it would seem important to compare RiboORF to the method as described here http://biorxiv.org/content/early/2015/11/13/031625 (in press in Nature Methods) which uses an alternative approach to use the 3nt pattern.

2) Yes I was referring to the multiple-testing that is the result of testing many possible ORFs. I did not see a false positive assessment which took into consideration what is discussed under point 1. The false positive calculations seem to be for the classifier only (Figure 2).

---

## [Author Response]

*Although the reviewers agree that you have addressed important aspects of the use of ribosome profiling to identify the full spectrum of the translatome, the major deficiency of this work is the validation of the data. Many reports have already documented ribosome footprinting of unexpected RNA species, but the next advance must be the demonstration that productive translation has indeed taken place. You state: "Future work including the affinity purification of 80S ribosomes and mass-spectrometry of small peptides will provide more accurate determination of the translational status of individual transcripts." However, ribosome affinity purification has been already applied (Ingolia et al., 2014; Zhou P et al., PNAS 2013) to show that the majority of noncoding RNAs, including most long intergenic noncoding RNAs, are ribosome-bound to the same extent as coding transcripts. You must by now have validation data to add to the paper.Ribosome profiling is a powerful technology that can be applied to identify ribosome protected positions in a genome wide unbiased fashion. You revisit one of the less explored aspects of this technique, namely to what extent ribosome profiling data can be used to identify true translation events in e.g. 5' UTRs and long non-coding RNAs. The core issue is that it is unknown to which extent non-ribosome related factors, scanning ribosomes etc. can also result in "ribosome-protected" fragments and to what extent such fragments can occur in a random fashion. You weigh in on this with a new, complementary approach to defining true translation in ribosome profiling data. You develop two tests, for codon periodicity and uniform coverage (as opposed to a single, high-abundance fragment). These approaches are different than the scoring metrics previously used by the Giraldez and Guttman groups, and this new approach is well validated here. You show that lncRNA translation tends to occur on transcripts with cytoplasmic (as opposed to nuclear) localization, which is a clear prediction of any model of lncRNA translation but has not previously been tested. You also develop several lines of evidence supporting protein-level conservation constraining a subset of translated lncRNA regions. Forty-one of these are conserved in mice, and represent candidate genes encoding tiny proteins. You argue for the translation of many pseudogenes, including continued selection on the protein-coding potential of these sequences.*General comments regarding validity and the polysome experiment:

I am surprised that the Reviewers do not accept the statement that “it is inconceivable the uniform 3-nt periodicity over an extended distance can result in anything other than bona fide translation”. Given that the majority of sequence reads are ribosome-protected fragments of a well-defined size and correspond to codons of canonical ORFs, I cannot imagine a better definition of translation. It is far superior to polysome-associated RNA, the long-time definition, which does not even map the translated region, much less have any connection to codons. Nevertheless, at the suggestion of the Reviewers, we perform a standard 80S/polysome experiment and show that all 7 RNAs tested that encode non-canonical, translated ORFs indeed are associated with 80S and/or polysomes (new Figure 2—figure supplement 2). Untranslated control RNAs are not associated with either the 80S or polysomes. The alternative explanations for the data mentioned in Review and discussed below are implausible; they are inconsistent with the very-well understood translation mechanism *and* the data.

1) Non-ribosomal RNA-protein complexes: Other than translating ribosomes, no known RNA-protein complexes have any relationship to codons, 3-nt periodicity, or extended regions. In fact, the opposite is true; such complexes protect very specific regions of RNA. Indeed, 11% of the sequencing reads are highly localized (low PME values), have no 3-nt periodicity, and are comparable in both cycloheximide- and harringtonine-treated cells. These are the non-ribosomal RNA-protein complexes (we have analyzed them in detail in other work).

2) Scanning ribosomes: Scanning ribosomes, which recognize the 5’ cap of RNA and scan down to the initiation codon, do not incorporate amino acids and do not recognize codons (except initiation codons). As such, scanning ribosomes do not result in 3-nt periodicity; if they did, it would be hard to understand how they select initiation codons independent of reading frame. More importantly, if one looks at ribosome profiles at classical ORFs, sequence reads start at the initiation codon. There are few if any reads upstream where the scanning ribosomes are. So, scanning ribosomes cannot possibly account for 3-nt periodicity, both in principle and according to the ribosome profiling data.

3) Random binding of ribosomes: There is no evidence that this occurs to any significant extent. Indeed, one of the key experiments leading to the scanning ribosome model was the inability of ribosomes to bind circular RNA (i.e. lacking an end). By the very definition, random binding would not occur in selected and extended regions with 3-nt periodicity.

4) Most general evidence for validity: The 3-nt periodicity for classical vs. non-conventional ORFs are indistinguishable; i.e. they all have the same relative% reads for the 1st, 2nd, and 3rd position. Furthermore, we showed that the translation efficiency (ribosome profiling reads: RNA seq reads) of non-canonical ORFs is comparable (only slightly less) than conventional ORFs. The Reviewers cannot seriously dispute that classical ORFs are translated. How then do they account for the indistinguishable 3-nt periodicity and roughly comparable translation efficiency of non-canonical ORFs? The idea that completely different mechanisms somehow yield remarkably similar results strains credulity.

1) What the field is in strong need of is a study where suggested translation events are validated at a large scale with an alternative approach than ribosome profiling. This could be mass spectrometry (there are some new approaches that identify and quantify ongoing protein synthesis events) and/or association with polysomes. Such a validation would allow for benchmarking the analysis approaches that are proposed. We believe that a reasonable number of validated targets would be 7-8, for example, from the forty-one that are conserved in mice, and represent candidate genes encoding tiny proteins.

As mentioned above, we performed the suggested polysome association experiments and obtained the expected results. We did not perform mass spectrometry to identify peptides, because this is not validation for translation. Peptide measurements don’t measure translation per se, because they also reflect stability of the protein. By analogy, transcription is measured by RNA polymerase (e.g. ChIP, Net-seq, GRO-seq), not RNA levels. It is well appreciated that RNA levels are a poor, and indeed unacceptable, approach for measuring transcription, because many “cryptic” RNAs are transcribed but unstable (except under particular conditions or in mutant strains).

2) Gerashchenko and Gladyshev,(2014 NAR) described a strong bias in ribosome profiling studies because of the use of cycloheximide, in particular affecting uORFs (but likely also long non-coding RNAs). It is surprising that the protocol used by the authors is not discussed in this context and it does seem possible that artifacts such as those described could indeed be a factor in the present study as well. It is unclear how the analysis approach described would deal with such artifacts.

I don’t understand the relevance of the Gerashchenko and Gladyshev paper. The cycloheximide artifacts in that paper related to stress conditions, which are not relevant here. They also were observed in yeast, not mammalian cells. Also, the main artifact related to a broad peak downstream of the start codon, and this is not observed in our data.

3) Pseudogenes retain substantial nucleotide-level identity with their protein-coding ancestors. Short ribosome footprints are particularly prone to mis-mapping, and the authors don't provide details on their handling of multi-mapping reads &c. They should exclude the possibility that apparent translation of pseudogenes is a result of ribosome footprints on conventional protein-coding ancestors.

Although it is true that “pseudogenes retain substantial nucleotide identity with their protein-coding ancestors”, there is no problem with distinguishing reads between these 2 classes of genes. The sequence reads are ~30 nt (paired end) and it is rare for nucleotide identity between pseudogenes and canonical genes to occur over this length. Moreover, we use standard Tophat parameters to eliminate non-unique reads. I should note that we have dealt with issue in other experiments (e.g. different tRNA genes; Moqtaderi et al, 2010 NSMB) where there was much more similarity between related genes.

4) Can the authors comment on why transcripts with ORFs of >100 (and in some cases >200) amino acids are nonetheless classified as lncRNAs?

This is a semantic issue, which we tried to address in the Discussion. There is no fixed boundary to distinguish between a “translated lncRNA” making a peptide >100 aa and an mRNA encoding a short peptide. Moreover, by definition, a translated lncRNA is not really a lncRNA, since it is codes for a peptide that is synthesized (although may not be stable). Perhaps one could make some kind of distinction based on how long the peptide is with respect to RNA length, but it is unclear if this has any meaning.

5) The uORF-mediated regulation of ATF4 in particular is well-studied by the Harding lab among others, and the authors should cite this literature in discussing these uORFs.

In addition to the appropriate paper already cited, we now cite a review article on ORF-mediated regulation of ATF4. We are well aware of this literature, and indeed cited it and mentioned our ATF4 result as confirmation of previous knowledge.

6) The authors say that, "By definition, noncoding RNAs should not be translated into protein" (in the Introduction) and "By definition, non-coding RNAs are not translated into protein" (in the Discussion). In fact, this is begging the question to some extent – there may be RNAs, e.g., that function as microRNA sponges in the cytosol but have a translated uORF whose translation is important only to avoid other translation that would interfere with this non-coding function (e.g. Ulitsky & Bartel).

For decades, “coding” has meant “an ORF/peptide sequence that is translated under some condition”. So, a non-coding RNA, by definition, is not translated. In the example mentioned, if an RNA functions as a miRNA sponge, that fact doesn’t bear on the question of whether this RNA is coding or non-coding. If this “sponge” RNA is translated, it is coding, no matter what the reason (if any) for why it is translated. Again, the last section of the Discussion attempted to deal with the semantic issues here. There are no simple remedies/terminologies for the many cases where RNAs are translated, but the peptides may be unstable and/or have no biological function. Perhaps the reviewers would suggest new terminology for this situation.

7) It is surprising that about 35% of all reads do not originate from the expected periodic position (Figure 1). This suggests that there is substantial randomness in the methodology, which would be expected to contribute to the stochastic characteristics of the data.

Figure 1 and other figures – The reason why 35% of all reads do not originate from the expected periodic position is simply because RNase does not perfectly (i.e. to the nucleotide) degrade unprotected RNA and fail to degrade protected RNA. This means alignment isn’t perfect, and no one would expect that it would be. More importantly, the 3-nt periodicity is striking by simple inspection, especially compared to control regions that are not translated but are bound by non-ribosomal RNA-protein complexes (Figure 2). The p-value for 3-nt periodicity (null hypothesis is non-ribosomal complexes) is 10-12, which is extremely compelling. And, this p-value is generated on a 10 aa peptide for an RNA expressed at the lower limit of our analysis; longer or more highly expressed peptides would have vanishingly smaller p-values. Of course, the RibORF algorithm specifically addresses the statistical significance of 3-nt periodicity for any putative ORF; that is the whole point and major advance of the method.

8) The approach presented relies on the 3 nt periodicity and random distribution (measured by entropy) of the reads across the ORF. From Figure 2 it appears that it is mainly the 3 nt periodicity that is driving the classification. Thus, one critical issue is to what extent such patterns can occur by chance under the multiple testing situations that are assessed. It is also unclear why we should assume that this situation cannot be the result of factors other than ribosomes or simply occur by chance when the authors states that "It is inconceivable that uniform 3nt period…".

Yes, 3-nt periodicity drives most, but not all, of the identification of translated ORFs. But, as discussed in point 6, the chance that the observed periodicity occurs by chance is small. It is unclear what is meant by “multiple testing situations”. If this means that there are many translated ORFs, then it is true that a very small number of ORFs may be false-positives. Indeed, we calculated the false-positive and false-negative rate in the paper, which are extremely low for any genome-scale analysis. This attests to the power of 3-nt periodicity. Lastly, the possibility that our results could be explained by “factors other than ribosomes or simply occur by chance” is inconsistent with current knowledge and the data.

9) The conservation analyses are in general relatively modest and it is hard to interpret whether this is a result of that there is a large fraction of false positive translation events and that the true ones are indeed conserved or whether there is an abundance of "unusual" translation events that are not conserved. Indeed many "peptides" are very short which would suggest a larger risk for false positive 3nt periodicity and uniform distribution of reads, especially for lowly expressed genes (it is not clear if there is bias for detecting more genes with low rpf counts as truly translated).

The conservation analyses are straightforward (though novel), and this comment is based on an incorrect premise. As indicated above, there is not a large fraction of false translation events, and all the identified translation events (even those as short as 10 aa) have 3-nt periodicity far above chance expectation. The Reviewers are correct that, like all other experiments of this general type, low-expressed genes are problematic and it might be difficult to establish statistical significance. However, the Reviewers may not have realized that RibORF explicitly deals with this issue, because there it involves a probabilistic cut-off to identify translation events. As such, we never identify translation events from poorly expressed RNAs where 3-nt periodicity is hard to see due to limited sequence reads. There is no bias for detecting more genes with low read counts; in fact, exactly the opposite, because more read counts means better statistics.

[Editors' note: further revisions were requested prior to acceptance, as described below.]

Reviewer #1:

1) The authors say that cytosolic lncRNAs are translated "invariably" in three places including in the Abstract. This word is too strong even for the set of lncRNAs present in this sample, and more broadly, the only invariable thing in biology is the presence of surprising exceptions.

The word “invariably” has been removed in all three places.

2) The authors don't seem to consider transcript isoform variation in their interpretation of uORFs and dORFs. In yeast, translated uORFs sometimes occur on small, independent transcripts (Arribere & Gilbert 2013, Pelechano & Steinmetz 2013) and so the lack of uORF-mediated repression may reflect the fact that the uORF is not translated from the same transcript as the CDS. Likewise, the highly translated dORFs may reflect translation of extensively 5'-truncated RNAs.

For most genes we examine, uORFs and dORFs are expressed in the same transcripts as canonical ORFs. However, it is possible for a few cases, uORF and dORFs are expressed in truncated RNAs, which we cannot assess in a genome-wide experiment. Such potential exceptions do not affect our general conclusions. We have added a sentence to this effect (paragraph two, subheading “RibORF identifies a large number of translated ORFs in lncRNAs, pseudogenes, and UTRs of mRNAs”).

Reviewer #2:

Regarding the comment on "it is inconceivable the uniform 3-nt periodicity over an extended distance can result in anything other than bona fide translation". This was mainly a concern regarding scientific style. For very few findings in science, if any, should alternative explanations be inconceivable. As discussed, 3-nt periodicity is very strong evidence for translation but as suggested by the authors it could also occur via other events and thus is not inconceivable at the single ORF level. This may not only include biological aspects but also the stochastic nature of data which will suggest 3-nt periodicity with some false positive rate. The stochastic nature of the data was the main concern of this reviewer.

We have removed the term “inconceivable” from the text (although I can’t conceive of an alternative model, and the Reviewer is incorrect that we suggested such alternatives; instead, we discussed why such alternatives were inconsistent with current knowledge and the data). I don’t understand the comment about “stochastic nature” of the data. Our use of the term “inconceivable” was meant conceptually; i.e. 3-nt periodicity over an extended region meant translation. Of course, data for an individual ORF has a probabilistic component, which is why there are false positives and false negatives at a given threshold, but this has nothing to do with the concept.

*1) The authors seem to agree that all the reads are not expected to be derived from RNA fragments that are protected by ribosomes. The issue that I have is that there is no background model for how this relatively large proportion of the reads would stochastically result in 3nt periodicity.In this context it would seem important to compare RiboORF to the method as described here*
http://biorxiv.org/content/early/2015/11/13/031625
*(in press in Nature Methods) which uses an alternative approach to use the 3nt pattern.*

The Reviewer appears to have missed our background model. Specifically, we used the internal off-frame ORFs of protein coding genes and ORFs in small noncoding RNAs as negative examples.

2) Yes I was referring to the multiple-testing that is the result of testing many possible ORFs. I did not see a false positive assessment which took into consideration what is discussed under point 1. The false positive calculations seem to be for the classifier only (Figure 2).

We presented the ROC curve in Figure 2 to show the performance of our algorithm. We mentioned the false discovery rates (FDR) in the manuscript, and these already account for multiple testing.